# Higher Order Equivariant Graph Neural Networks for Charge Density Prediction

**Teddy Koker**       **Keegan Quigley**       **Lin Li**
MIT Lincoln Laboratory
Lexington, MA 02421
{thomas.koker,keegan.quigley,lin.li}@ll.mit.edu

## Abstract

The calculation of electron density distribution in materials and molecules is central to the study of their quantum and macro-scale properties, yet accurate and efficient calculation remains a long-standing challenge in the field of material science. This work introduces ChargE3Net, an E(3)-equivariant graph neural network for predicting electron density in atomic systems. Unlike existing methods, ChargE3Net achieves equivariance through the use of higher-order tensor representations, and directly predicts the charge density at a set of desired locations. We demonstrate the effectiveness of ChargE3Net on large and diverse sets of molecules and materials, where it achieves state-of-the-art performance over existing methods, and scales to larger systems than what is feasible to compute with density functional theory. Through additional experimentation, we demonstrate the effect of introducing higher-order equivariant representations, and why they yield performance improvements in the charge density prediction setting.

## 1  Introduction

Electronic charge density is one of the most fundamental quantities in quantum chemistry and physics and it is key to accurately modeling molecules and materials at the atomic scale. Hohenburg-Kohn theorem [1] states that the ground-state charge density contains the information necessary to obtain all ground-state properties of interest. Density functional theory (DFT) that solves the Kohn-Sham (KS) equations is the most widely used *ab initio* method for performing electronic structure calculation of molecules or materials. However, KS DFT is computationally expensive with $O(N^3)$ complexity, making it infeasible for large-scale quantum calculations.

More recently, machine learning models have been developed to overcome the computational challenge of *ab initio* calculations. Early approaches using machine learning for the prediction of electron density were based on the use of symmetry-adapted Gaussian process regression to predict coefficients for atom-centered basis functions [2, 3]. Coefficients are predicted from a kernel function that expresses the structural similarity and geometric relationship among a target atomic environment and a training set of atomic environments. More recent work has leveraged invariant and equivariant neural networks to predict these coefficients from atomic features. Qiao et al. [4] uses mean-field electronic structure, computed from the GFN-xTB [5] quantum mechanical model, as inputs for a higher-order equivariant neural network to predict basis set coefficients. Rackers et al. [6] use a higher-order equivariant neural network to predict basis set coefficients directly from atomic graphs. While these approaches are able to achieve high accuracy in some settings, they are limited by the expressivity of density fitting basis sets. These atom-centered basis sets must be hand selected, and are often not sufficient for periodic systems where plane wave basis functions are more appropriate.

Alternatively, several methods were proposed to learn electron density directly from a discretized grid of density points. By inserting each grid or "probe" point into the atomic graph, charge density

prediction can be modeled as a node regression task [7–9]. Gong et al. [7] achieve this through the use of an invariant graph convolutional network [10] on a small dataset of crystalline polymers, demonstrating transferability to unseen structures. Jørgensen and Bhowmik [8] similarly use an invariant graph convolutional network called SchNet [11], demonstrating fast and high accuracy charge density prediction in molecules, liquids and inorganic solids. They later demonstrate an improvement in accuracy through the use of an equivariant graph convolutional network called PaiNN [9], through $\mathbb{R}^3$ vector representations and rotationally equivariant operations [12]. While these models have been shown to achieve high accuracy when trained and evaluated on small, specialized molecular or material datasets, their expressive power on larger, diverse datasets such as the Materials Project [13] is not yet understood. Furthermore, these models use scalar and/or vector representations to construct rotationally equivariant networks, which could be further improved by incorporating higher-order equivariant features, as has been shown in other tasks, such as the prediction of atomic forces [14].

In this work we introduce ChargE3Net, a higher-order equivariant graph neural network for predicting the electron density of atomistic systems. Our method is purely data driven, and achieves equivariance through higher order tensor representations and SO(3) equivariant operations. Despite using only atomic graphs as input, our proposed method achieves state-of-the-art accuracy in the prediction of charge density in both materials and organic molecules, outperforming methods relying on features obtained through quantum mechanical simulation. We study the effect of introducing higher rotation order representations to our model, validating that the addition of higher order representations results in improvements in accuracy. We exemplify this further by quantifying the angular dependence of the charge density with respect to atom positions, experimentally demonstrating the need for higher rotation order representations to model high angularly dependent systems. Lastly, we demonstrate the linear time complexity of our model with respect to system size, showing the capability of predicting density on systems with $> 10^4$ atoms, surpassing what is feasible with *ab initio* calculations.

## 2 Methodology

### 2.1 Equivariance

Our model seeks to predict charge density $\rho(\vec{r}) : \mathbb{R}^3 \to \mathbb{R}$ for a given atomic system. The charge density, along with other properties such as forces, is equivariant under rotation and translation of the atomic system. In other words, a rotation or translation of the atomic system in Euclidean space will result in an equivalent rotation or translation to the charge density or forces. Formally this is known as equivariance with respect to E(3), which includes rotations, translations, and reflections in 3D space. This can be achieved in part by using only *invariant* scalar features [11], such as inter-atomic distances; however, this prevents the model from using angular information, limiting the accuracy that can be attained [15]. More recent work [12] has introduced equivariance methods using vector $\mathbb{R}^3$ features, such as relative atomic positions, to incorporate angular information, and has been shown to improve the performance of charge density prediction [9].

Our work achieves higher-order equivariance through means outlined by tensor field networks [16]. Translation equivariance is achieved by using relative atomic positions, and rotation equivariance is achieved by restricting features to be irreducible representations, *irreps*, of SO(3). These features take the form $V_{cm}^{(\ell,p)}$, a dictionary of tensors with keys representing rotation order $\ell \in \{0, 1, 2, ...\}$ and parity $p \in \{-1, 1\}$. Each tensor has a channel index $c \in [0, N_{\text{channels}})$ and an index $m \in [-\ell, \ell]$. In this way, the representation at a given $\ell$ and $p$ will have a size of $\mathbb{R}^{N_{\text{channels}} \times (2\ell+1)}$. These representations are combined with the tensor product operation $\otimes$, using Clebsch–Gordan coefficients $C$ as described in [16] and implemented in e3nn [17]:

$$\left( U^{(\ell_1,p_1)} \otimes V^{(\ell_2,p_2)} \right)_{cm_o}^{(\ell_o,p_o)} = \sum_{m_1=-\ell_1}^{\ell_1} \sum_{m_2=-\ell_2}^{\ell_2} C_{(\ell_1,m_1)(\ell_2,m_2)}^{(\ell_o,m_o)} U_{cm_1}^{(\ell_1,p_1)} V_{cm_2}^{(\ell_1,p_2)} \tag{1}$$

where $\ell_o$ and $p_o$ are given by $|\ell_1 - \ell_2| \leq \ell_o \leq |\ell_1 + \ell_2|$ and $p_o = p_1 p_2$. We maintain only those representations with $\ell_o \leq L$ where $L$ is a maximum allowed rotation order. For ease of notation, we will omit the keys $\ell$, $p$ and indices $c$, $m$ for these tensor representations for the rest of the paper.

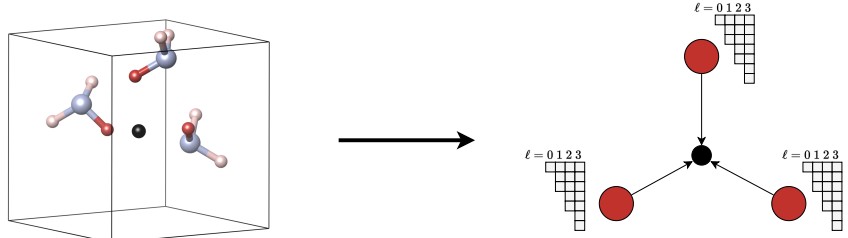

Figure 1: Illustration of single charge density probe point. shown in black, in a periodic atomic system. A graph is formed through neighboring atoms, and messages composed of scalars ($\ell = 0$), vectors ($\ell = 1$) and higher-order tensors ($\ell \geq 2$) are aggregated at probe point vertices.

## 2.2 Architecture

We represent predicted charge density $\hat{\rho}(\vec{g})$ as a neural network with inputs of atomic numbers $\{z_1, ..., z_N\} \in \mathbb{N}$ with respective locations of the atoms $\{\vec{r}_1, ..., \vec{r}_N\} \in \mathbb{R}^3$, as well as probe locations $\{\vec{g}_1, ..., \vec{g}_M\} \in \mathbb{R}^3$ where charge densities are to be predicted, and an optional periodic boundary cell $B \in \mathbb{R}^{3\times3}$ for periodic systems.

As illustrated in Figure 1, a graph is constructed with atoms and probe points as vertices, with edges formed via proximity with a cutoff of 4 Å. Our graph neural network is formulated such that message passing between atom and probes is unidirectional, with probe point only receiving messages. Through each layer $n$ of the network, atoms and probe points will maintain tensor representations $A^n$ and $P^n$ respectively (section 2.1). $A^n$ is initialized as a one-hot encoded $z$, represented as a $\ell = 0$, $p = 1$ tensor with $N_{\text{channels}}$ equal to the number of atomic species. $P^n$ is initialized as single scalar zero, with $\ell = 0$, $p = 1$, and $N_{\text{channels}} = 1$. Each representation is updated through a series of alternating convolution $\text{Conv}(\cdot)$ and non-linearity $\text{Gate}(\cdot)$ layers. Atom representations are updated with $A_i^{n+1} = \text{Gate}(\text{Conv}_{\text{atom}}^n(\vec{r}_i, A_i^n))$. $\text{Conv}_{\text{atom}}$ is defined as:

$$\text{Conv}_{\text{atom}}^n(\vec{r}_i, A_i^n) = W_1^n \left( \sum_{j \in N(i)} W_2^n A_j^n \otimes R(r_{ij}) Y(\hat{r}_{ij}) \right) + W_3^n A_i^n \tag{2}$$

where $W_1, W_2, W_3$ are learned weights applied as a linear mix or self-interaction [16]. The set $N(i)$ includes all atoms within the cutoff distance, including those outside potential periodic boundaries. $r_{ij}$ is the distance from $\vec{r}_i$ and $\vec{r}_j$, with unit vector $\hat{r}_{ij}$. $Y(\hat{r}_{ij})$ are spherical harmonics, and $R(r_{ij}) \in \mathbb{R}^{N_{\text{basis}}}$ is a learned radial basis function defined as:

$$R(r_{ij}) = \text{MLP}([\phi_1(r_{ij}), ..., \phi_{N_{\text{basis}}}(r_{ij})]) \tag{3}$$

where $\text{MLP}(\cdot)$ is a two layer multilayer perceptron with SiLU non-linearity [18], $\phi(\cdot)$ is a gaussian basis function $\phi(r_{ij})_k \propto \exp(-(r_{ij} - \mu_k)^2)$ with $\mu_k$ uniformly sampled between zero and the cutoff, then normalized to have a zero mean and unit variance.

The convolution updates the representation of each atom to be the sum of tensor products between neighboring atom representations and the corresponding spherical harmonics describing their relative angles, weighted by a learned radial representation. This sum is followed by an additional self-connection, then a residual self-connection. The output of the convolution is then passed though an equivariant non-linearity $\text{Gate}(\cdot)$ operation as described in [19]. We use SiLU and $\tanh$ activation functions for even and odd parity scalars respectively, as is done in [14].

Probe representations are updated similarly as the atoms for each layer, except their representations depend solely on the representations of neighboring atoms, with no probe-probe interaction. Each

probe representation is updated with $P_k^{n+1} = \text{Gate}(\text{Conv}_{\text{probe}}^n(\vec{g}_k, P_k^n))$, where $\text{Conv}_{\text{probe}}$ is defined as:

$$\text{Conv}_{\text{probe}}^n(\vec{g}_k, P_k^n) = W_1^n \left( \sum_{i \in N(k)} W_2^n A_i^n \otimes R(r_{ik}) Y(\hat{r}_{ik}) \right) + W_3^n P_k^n \quad . \tag{4}$$

Note that weights $W$ are not shared with those for $\text{Conv}_{\text{atom}}^n(\cdot)$. Since atom representations are computed independently of probe positions, they can be computed once per atomic configuration, even if multiple batches are required to obtain predictions for all probe points. Finally, predicted charge density $\hat{\rho}(\vec{g}_k)$ is computed as a linear combination of the scalar elements of the final representation $P_k^{n=N_{\text{layers}}}$.

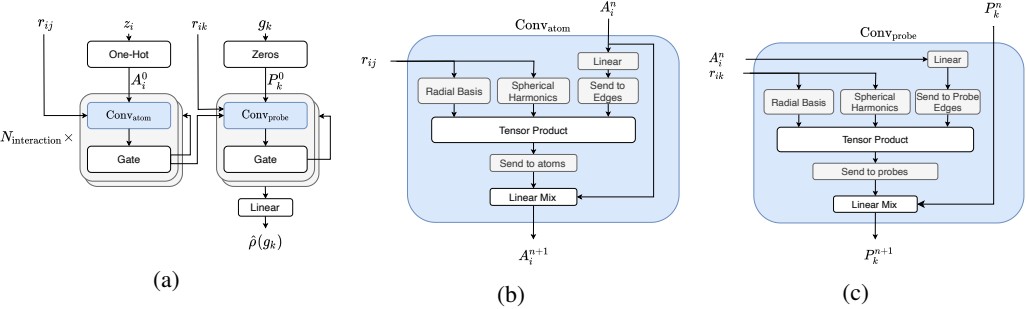

Figure 2: a: Overall ChargE3Net architecture. Predicted charge density $\hat{\rho}(\vec{g}_k)$ at probe point $\vec{g}_k$ is computed from atomic species $z_i$, interatomic displacement vectors $r_{ij}$, and probe-atom displacement vectors $r_{ik}$. b: Atom graph convolution. c: Atom-probe graph convolution.

## 3 Experiments

We train and evaluate our models on VASP [20] computations of the organic molecules within the QM9 dataset [21–23], mixed transition metal layered oxide lithium ion battery cathode materials (NMC) [24], and inorganic materials collected from Materials Project (MP) [13, 25]. The QM9 and NMC datasets are provided by Jørgensen and Bhowmik [9]. For the MP data, we collected 122,689 structures and associated charge densities from `api.materialsproject.org`[1]. Structures in the dataset that shared composition and space group were identified as duplicates and only the highest `material_id` structure was included, leaving 108,683 materials. The data was split into training, validation, and test splits with sizes 106,171, 512, and 2000 respectively. 26 materials in the test set were later found to have unrealistic volume per atom, in excess of 100 $\text{Å}^3$/atom, and these were removed from the test set and excluded from results.

Table 1 outlines the experimental setup for training ChargE3Net on each of the datasets. For each gradient step, a random batch of materials is selected, from which a subset of the charge density probe points are used. We use the Adam optimizer [26], and decay the learning rate by $0.96^{\beta s}$ at step $s$. We find that optimizing for L1 error improves performance and training stability over mean squared error.

Table 1: Training setup

| Dataset | Learning Rate | Decay $\beta$ | $L$ | Batch Size | Training Steps |
|---|---|---|---|---|---|
| NMC | 0.01 | $10^4$ | 4 | 8 * 200 points | $7.5 * 10^5$ |
| QM9 | 0.01 | $10^4$ | 4 | 8 * 200 points | $10^6$ |
| Materials Project | 0.005 | $3 * 10^3$ | 4 | 16 * 200 points | $10^6$ |

---

[1]Collected 7 May 2023. Associated task identifiers will be included in our provided repository, along with train, validation, and test splits.

Following recent work [2–4, 9] we evaluate our models using mean absolute error normalized by the total number of electrons in the volume, $\epsilon_{\text{mae}}$ (Eq. 5). This is approximated via numerical integration on the charge density grid created by VASP.

$$\epsilon_{\text{mae}} = \frac{\int_{\vec{r} \in V} |\rho(\vec{r}) - \hat{\rho}(\vec{r})|}{\int_{\vec{r} \in V} |\rho(\vec{r})|} \tag{5}$$

We compare the performance of ChargE3Net to the models introduced in DeepDFT[9] and OrbNet-Equi[4]. For QM9 and NMC datasets we use identical training, validation, and test splits as Jørgensen and Bhowmik [9], and report $\epsilon_{\text{mae}}$ computed using the authors publicly released models. We have verified these to match the numbers reported in the original work. For the MP dataset, we train the DeepDFT models using the same experimental settings from the original work. As shown in Table 2, our model significantly outperforms the prior equivariant DeepDFT models [9] on the Materials Project and QM9 datasets, while achieving similar performance on the NMC dataset. In addition, our model achieves a lower $\epsilon_{\text{mae}}$ on the QM9 dataset than OrbNet-Equi [4] model, which leverages additional features based on quantum mechanical calculations, despite ChargE3Net only using atomic species and position information.

Table 2: Error for each dataset, reported in average $\epsilon_{\text{mae}}$ (%), $\pm$ one standard error.

| Dataset | invDeepDFT | equiDeepDFT | OrbNet-Equi | ChargE3Net (Ours) |
|---|---|---|---|---|
| NMC | $0.089 \pm 0.001$ | $\mathbf{0.061 \pm 0.001}$ | - | $\mathbf{0.060 \pm 0.001}$ |
| QM9 | $0.357 \pm 0.001$ | $0.284 \pm 0.001$ | $0.206 \pm 0.001$ | $\mathbf{0.196 \pm 0.001}$ |
| Materials Project | $0.859 \pm 0.011$ | $0.799 \pm 0.010$ | - | $\mathbf{0.523 \pm 0.010}$ |

## 3.1 Effects of Higher Order Representations

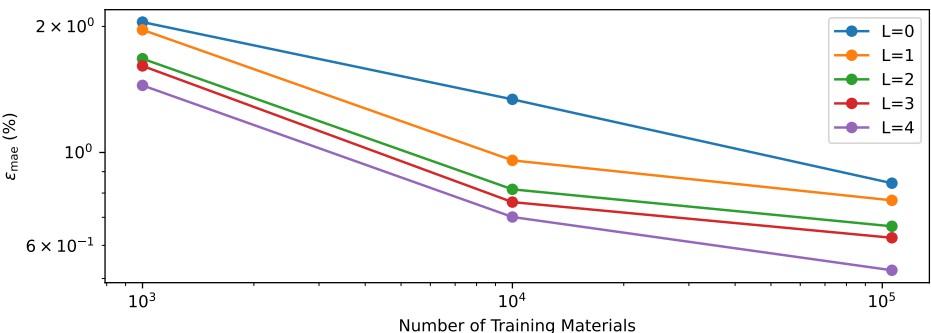

Figure 3: Log-log plot of training set size vs performance, measured in average $\epsilon_{\text{mae}}$ (%) on the Materials Project test set. We show models train with maximum rotation order $L \in 0, 1, 2, 3, 4$.

To demonstrate the impact of introducing higher order tensor representations in our model, we examine the effect of training models while varying the maximum rotation order $L$, with $\ell \in \{0, ..., L\}$ up to $L = 4$. While higher order representations are achievable, they can be prohibitive to use in the network due to the $O(L^6)$ computational complexity of tensor products. The number of channels for each order is determined by $N_{\text{channels}} = \lfloor \frac{500/(L+1)}{\ell * 2 + 1} \rfloor$. For example, the $L = 0$ model has representations consisting of 500 even scalars, and 500 odd scalars, while the $L = 1$ model has representations consisting of 250 even scalars, 250 odd scalars, 83 even vectors, and 83 odd vectors. In this way, each model is constructed such that the total representation size is approximately equal, as well as the proportion of the total representation used by each order. We train each model on 1,000 and 10,000 material subsets of the MP dataset as well as the full dataset, using the same validation and test sets. Figure 3 shows a consistent increase in performance on each subset with the addition of each rotation order and similar scaling behavior with respect to the training set size. This trend suggests that higher order representations are necessary for accurate charge density modeling, and can match the performance of lower order models with less data.

In order to gain intuition behind why and when higher order representations yield higher performance, we consider two factors contributing to the variance of charge density within a material: *radial* dependence, or a dependence on the distance from neighboring atoms, and *angular* dependence, a dependence on the angle from a point with respect to the rest of the system. While most materials exhibit strong radial dependence, some also exhibit strong angular dependence. This would be difficult to model using an invariant architecture ($\ell = 0$) operating solely on interatomic distances, and likely would require higher order representations to model correctly. In Figure 4, we illustrate this concept with two materials. $H_4Cs_2O_8P_2$ exhibits high angular dependence, where charge density is dependent on angle as well as radial distance from the nearest atom. Conversely, $Rb_2Sn_6$ does not exhibit this, as its density appears to be almost entirely a function of the atomic distance, suggesting that a lower-order equivariant or invariant architecture could model its electron density well. We find this intuition to be consistent with model performance, as Figure 4 shows similar performance for $L = 0$ and $L = 4$ models for $Rb_2Sn_6$, while the $L = 4$ model exceeds the performance of the $L = 0$ model for $H_4Cs_2O_8P_2$.

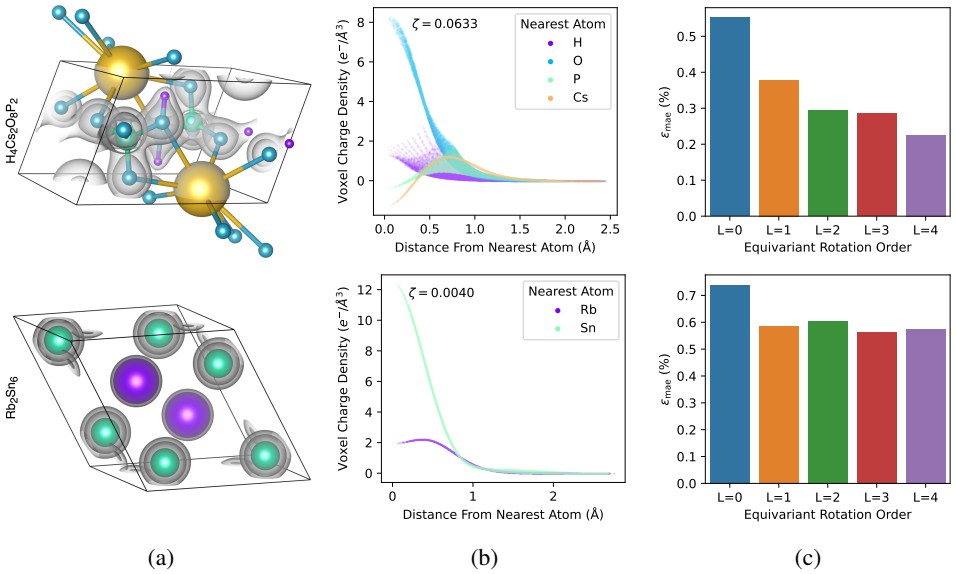

(a)           (b)           (c)

Figure 4: Comparison of material with high angular variance $H_4Cs_2O_8P_2$ (top) and low angular variance $Rb_2Sn_6$ (bottom), as determined by 95th- and 5th-percentile $\zeta$ values in test set. a: visualization of charge density isosurfaces (gray). b: plot of charge density with respect to radial distance from nearest atom. c: $\epsilon_{\text{mae}}$ for ChargE3Net model predictions on these materials.

To further quantify this angular dependence, we develop a metric $\zeta$ to determine to what extent an atomic system exhibits more angular variation in its charge density with respect to atom locations. This is achieved by measuring the dot product of the unit vector between a probe point and its nearest neighboring atom and the gradient of the density at that probe point:

$$\zeta(G) = 1 - \frac{\sum_{\vec{g}_k \in G} |\nabla \rho(\vec{g}_k) \cdot \hat{r}_{ki}|}{\sum_{\vec{g}_k \in G} ||\nabla \rho(\vec{g}_k)||} \tag{6}$$

where $G$ is a set of probe points and $\hat{r}_{ki}$ is a unit vector from probe point at location $\vec{g}_k$ to the closest atom at location $\vec{r}_i$. Intuitively, a material with charge density gradients pointing directly towards or away from the nearest atom will have dot products larger in magnitude and $\zeta \rightarrow 0$, whereas a material with density that changes angularly with respect to the nearest atom will have dot products smaller in magnitude, and $\zeta \rightarrow 1$. Figure 5 demonstrates that the differences in performance between the lower and higher rotation order networks correlates strongly with $\zeta$. As electron density distributions with higher angular variance do occur naturally in the data, this further justifies the need for introducing higher rotation order representations into charge density prediction networks.

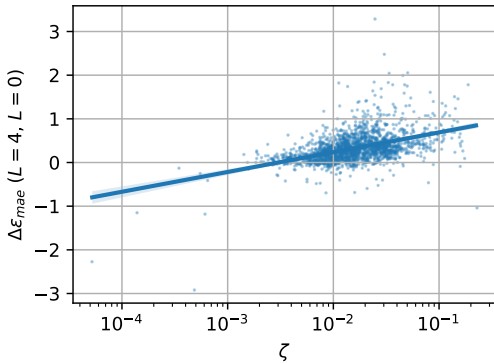

Figure 5: Scatter of $\epsilon_{mae}$ improvement from $L = 0$ to $L = 4$ vs. $\zeta$, showing greater performance gains from the higher order model when materials exhibit high angular variance.

## 3.2 Runtime

To demonstrate the scalability of our model, we analyze the runtime duration of our model on systems with increasing number of atoms, and compare to the duration of DFT calculations. We run each method on a single material, $BaVS_3$ (mp-3451 in MP), creating supercells from $1 \times 1 \times 1$ to $10 \times 10 \times 10$ and recording the runtime to generate charge density on the system. Figure 6 shows an approximate linear, $O(N)$, scaling of our model with respect to number of atoms, while DFT exhibits approximately $O(N^{2.3})$ before exceeding our computational budget. This is to be expected, as the graph size increases linearly with cell volume if the resolution remains the same, while DFT has shown to scale at a cubic rate with respect to system size [9]. Furthermore, like DeepDFT, our method can be fully parallelized up to each point in the density grid with no communication overhead.

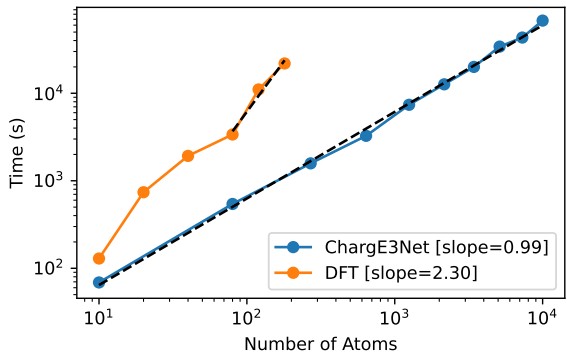

Figure 6: Runtime comparison of DFT and our ChargE3Net model with respect to number of atoms in the evaluated system. DFT is using a 48-core Intel Xeon CPU, while ChargE3Net uses a single NVIDIA V100 GPU.

## 4 Discussion

This work introduces an architecture for predicting charge density through equivariance with higher order representations. We demonstrate that introducing higher order tensor representations over scalars and vectors used in prior work achieves greater predictive accuracy, and show how this achieved through the improved modelling of systems with high angular variance. While our models use up to $L = 4$ representations, they generally use very few channels due to the weighting strategy mentioned in Section 3.1. As tensor products have a computation complexity of $O(L^6)$, there may be a more optimal model configuration with a uniform distribution of channels at a lesser $L$. Furthermore, SO(3) convolutions can be approximated in SO(2) [27] which brings down the computational complexity to $O(L^3)$ and removes the need to compute Clebsch-Gordan coefficients.

Future work may investigate these optimization and efficiency improvements, as well as the integration of charge density prediction models to existing DFT frameworks for downstream property prediction and simulation.

## Acknowledgments and Disclosure of Funding

DISTRIBUTION STATEMENT A. Approved for public release. Distribution is unlimited. This material is based upon work supported by the Under Secretary of Defense for Research and Engineering under Air Force Contract No. FA8702-15-D-0001. Any opinions, findings, conclusions or recommendations expressed in this material are those of the author(s) and do not necessarily reflect the views of the Under Secretary of Defense for Research and Engineering. © 2023 Massachusetts Institute of Technology. Delivered to the U.S. Government with Unlimited Rights, as defined in DFARS Part 252.227-7013 or 7014 (Feb 2014). Notwithstanding any copyright notice, U.S. Government rights in this work are defined by DFARS 252.227-7013 or DFARS 252.227-7014 as detailed above. Use of this work other than as specifically authorized by the U.S. Government may violate any copyrights that exist in this work.

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
