# OpenReview forum: "Higher Order Equivariant Graph Neural Networks for Charge Density Prediction"
_NeurIPS.cc/2023/Workshop/AI4Science — NeurIPS2023-AI4Science Poster_

### Official Review · Reviewer_m3QE · 2023-10-22
**Review of 'Higher Order Equivariant Graph Neural Networks for Charge Density Prediction'**

**Rating:** 9
**Confidence:** 3

**Review:**

The authors in this paper propose an equivariant GNNs, via high-order tensor representations, in order to predict the charge-density of molecules and materials. Overall, the paper is well-written, and the discussions and descriptions on the methodology and results are comprehensive. The model has been tested on three datasets, indicating better accuracy than baseline models. The effects of high-order representations have been discussed, and the authors further analyzed the results by introducing a new metric quantifying the angular dependence. Lastly, a better scaling is shown compared with DFT method.

Some small doubts: (1) in the paper it was mentioned that the training data was generated via VASP software, but QM9 data was actually generated by Gaussian according the reference. Is that a typo? (2) In the benchmark results in Table 2, can the author confirm if the testing is evaluated on the same training-validation-testing split? Did the authors run the evaluations on baseline models by themselves, or cited the results from references?

---

### Meta-Review · Program_Chairs · 2023-10-26

**Recommendation:** Accept (Poster)
**Confidence:** 3

**Metareview:**

The paper presents ChargE3Net, to the challenge of predicting electron density distribution in materials and molecules. This method utilizes an E(3)-equivariant graph neural network and distinguishes itself by leveraging higher-order tensor representations to maintain equivariance and directly predict charge density at specific locations.

The reviewer appreciates the comprehensiveness of the paper in both its methodological discussions and results. The benchmarks indicate that ChargE3Net consistently outperforms the baselines. The reviewer acknowledges the depth the authors went into by discussing the effects of high-order representations and the introduction of a novel metric quantifying angular dependence. They also appreciate the work's analysis with the DFT method in terms of scalability.

However, two concerns are raised:

- A potential error regarding the software used for generating training data.
- A clarification is sought on whether the same training-validation-testing split was utilized for the results in Table 2 and if the authors independently evaluated the baseline models.

Considering the paper's high-quality insights, relevance to the workshop's theme, and its potential to inspire discussions and community building, I concur with the reviewer's decision and recommend it for an oral presentation